# Modeling Postmortem Ethanol Production/Insights into the Origin of Higher Alcohols

**DOI:** 10.3390/molecules27030700

**Published:** 2022-01-21

**Authors:** Vassiliki A. Boumba

**Affiliations:** Department of Forensic Medicine & Toxicology, Faculty of Medicine, School of Health Sciences, University of Ioannina, 45110 Ioannina, Greece; vboumba@uoi.gr or vassiliki.boumba@gmail.com; Tel.: +30-26-5100-7724

**Keywords:** ethanol analysis, blood alcohol concentration (BAC), higher alcohol, postmortem, congener alcohol, modeling, biomarker

## Abstract

The forensic toxicologist is challenged to provide scientific evidence to distinguish the source of ethanol (antemortem ingestion or microbial production) determined in the postmortem blood and to properly interpret the relevant blood alcohol concentration (BAC) results, in regard to ethanol levels at death and subsequent behavioral impairment of the person at the time of death. Higher alcohols (1-propanol, 1-butanol, isobutanol, 2-methyl-1-butanol (isoamyl-alcohol), and 3-methyl-2-butanol (amyl-alcohol)) are among the volatile compounds that are often detected in postmortem specimens and have been correlated with putrefaction and microbial activity. This brief review investigates the role of the higher alcohols as biomarkers of postmortem, microbial ethanol production, notably, regarding the modeling of postmortem ethanol production. Main conclusions of this contribution are, firstly, that the higher alcohols are qualitative and quantitative indicators of microbial ethanol production, and, secondly that the respective models of microbial ethanol production are tools offering additional data to interpret properly the origin of the ethanol concentrations measured in postmortem cases. More studies are needed to clarify current uncertainties about the origin of higher alcohols in postmortem specimens.

## 1. Introduction

Blood ethanol analysis is the most frequent test performed in a forensic toxicology laboratory, as part of the investigation process of death, accident, or crime. The applied analytical procedures are rather simple and provide accurate, specific, and precise results on ethanol concentration. The relative analytical data are used to support the forensic evidence, as part of the judicial inquiry, in relation to the inter-relation between the consumption of alcoholic beverages, the measured blood ethanol concentration, and the impairment of body functions [1,2,3]. Particularly, in postmortem cases, ethanol concentration must be determined, as part of death investigation since it might be a causal or a contributory factor. The determined postmortem ethanol concentration needs to be interpreted accurately, regarding its origin, exogenous or endogenous, or whether the measured blood alcohol concentration (BAC) has exceeded a legal threshold limit [1].

The possible sources of ethanol detected in postmortem specimens could be either antemortem ingestion by a living person who consumed alcoholic beverages, or antemortem endogenous production due to microbial fermentation in the intestine (“auto brewery” syndrome [4]), or postmortem microbial neoformation, either in corpse pre-sampling or in situ post-sampling [2,3]. However, the microbial-induced postmortem ethanol formation, as well as the source and integrity of the selected samples, is the major determinant that could complicate the interpretation of results [2].

Therefore, the discrimination of the source of ethanol, ante mortem ingestion or postmortem microbial production, is of vital importance for the proper interpretation of postmortem BAC results. It is the forensic toxicologist who is challenged to provide the relative, available scientific evidence on the ethanol levels at death and the subsequent behavioral impairment of the person at the time of death [2,3,5,6].

Factors, such as the putrefaction state of the cadaver at autopsy, the clinical history of the deceased, the determination of glucose levels, the identification of microbes in the analyzed sample, and the evaluation of the discrepancies between ethanol concentration from various sampling sites and from different specimens, have been used to evaluate the origin of the measured ethanol, in the effort to achieve feasible accuracy in interpreting the postmortem ethanol analysis results [1,2,3,6]. Furthermore, several biochemical molecules have been suggested to be determined in the biological specimens, and to be evaluated, mainly, as qualitative indicators of either antemortem ethanol consumption [7,8,9,10,11,12,13,14,15], or postmortem neoformation of ethanol [16,17,18,19,20,21,22,23,24,25,26].

The detection of low molecular weight volatiles in human specimens has been related, for decades, with microbial activity and ethanol neoformation, while their presence in postmortem specimens has been correlated with putrefaction and microbial activity [27]. Among the volatile compounds that are often detected in postmortem specimens are included the so-called higher alcohols [2]. The term “higher alcohols” is used in the literature to define mainly 1-propanol (n-propanol), isobutanol (methyl-1-propanol), 2-methyl-1-butanol (isoamyl-alcohol), 3-methyl-2-butanol (amyl-alcohol), and 1-butanol (n-butanol). They are also included among the “congener alcohols” that are determined during alcohol congener analysis of alcoholic beverages or blood [2,25]. Moreover, the term “branched chain alcohols” is used to define isobutanol, isoamyl, and amyl alcohol since they result from the branched-chain amino acids valine, leucine, and isoleucine, respectively. Among higher alcohols, 1-propanol has attracted the most attention [2,15,16,19,26,28,29], and 1-butanol has been directly related to putrefaction [30,31,32].

Relatively recently, a quantitative relationship between the microbially produced ethanol and the higher alcohols was described and was characterized as “modeling postmortem ethanol production” [33,34,35,36]. The respective studies reported the process for the bacteria species *E. coli* [33,34], *K. pneumoniae* [35], *S. aureus* [35], and *E. faecalis* [35]; the clostridia species *C. perfringens* [33] and *C. sporogenes* [33]; and the yeast specie *C.albicans* [36]. The constructed models were suggested as quantitative tools to estimate the microbially produced ethanol at the postmortem.

This contribution is a brief review of, mainly, the recent available literature aiming to investigate the role of the higher alcohols as biomarkers of postmortem, microbial ethanol production, notably regarding the modeling of postmortem ethanol production.

## 2. Microbial Generation and Detection of Higher Alcohols

### 2.1. Fermentation Pathways Generating Ethanol and Higher Alcohols 

Human body decomposition is a complicated process that differs among cadavers, and even between different parts of the same body, and it is affected by environmental conditions [37]. Shortly after death, microbes of the intestine (or the respiratory system and the environment) invade body tissues of the deceased, grow, and, as the available oxygen diminishes, they adapt their growth to anaerobic conditions (if possible for them), or, they give succession to the growth of anaerobic populations. Meanwhile, the composition of the available substrates in the cadaver changes gradually: glucose and other monosaccharides are depleted; and polymers (lipids, carbohydrates, and proteins) are converted to their monomers (fatty acids, glycerol, monosaccharides, amino acids). The microbial metabolism results in the neoformation of ethanol and higher alcohols, as well as other fermentation products (reported in detail previously [38]).

Bacteria and yeasts produce ethanol from the fermentative breakdown of carbohydrates, especially hexoses, such as glucose, which is the preferred carbon and energy source for many microbes. The most used pathway for glucose degradation is the Embden-Meyerhof-Parnas (EMD) glycolytic pathway. The sum of the reactions generates two molecules of ethanol from a glucose molecule, which, in other words (when molecules are converted to masses), means that glucose fermentation produces ethanol with half mass of the fermented glucose. In addition, enterobacteria could ferment sugars through the mixed acid and butanediol fermentation. Finally, ethanol could be produced from glycerol (a product of lipid hydrolysis or carbohydrate metabolism) by enterobacteria, clostridia, and yeasts (Table 1). This latter pathway results in the simultaneous neoformation of 1-butanol. Another biochemical pathway to produce 1-butanol from carbohydrates is the butyrate-butanol-acetate fermentation, which proceeds with the parallel formation of ethanol (among other products) and is followed by clostridia (Table 1).

The “branched chain alcohols” and 1-propanol result from the respective alpha-ketoacids, through the “Ehrlich pathway”. The higher alcohol 1-propanol could be a product of other pathways that ferment glycerol or acetone, and which proceed in parallel with the “Ehrlich pathway” and the ethanol production. Acetone could be a final product or an intermediate of the metabolism of carbohydrates and glycerol, or the catabolism of fatty acids (Table 1). This association of the microbial 1-propanol neoformation to glucose fermentative pathways and to the fermentation of other biochemical substrates could explain why this higher alcohol is the most abundant alcohol detected in the postmortem stage besides ethanol.

### 2.2. Detection of Higher Alcohols

The most applied analytical methodology to detect ethanol and higher alcohols, among other congeners, in biological specimens and alcoholic beverages, is gas chromatography (GC) engaged to flame ionization detector (FID), using either direct or, most often, head space (HS) sampling [2,3,25]. The GC-FID methods could be used, both for qualitative and quantitative analysis, by the use of appropriate internal standards, such as tert-butanol or acetonitrile [3,33,35]. The applied GC techniques require minimal sample handling and are the same for clinical or autopsy samples. The reported methods are specific and sensitive, allowing baseline chromatographic separation of the volatiles of interest, in relatively short analysis time. Limits of detection at the level of 0.1 mg/dL for ethanol, and 0.005 mg/dL for higher alcohols, have been achieved [35]. Reported precision of the routinely blood alcohol analysis methods is high, with inter-laboratory coefficients of variation (CV) at 3–5%, and intra-laboratory CVs less than 1% [3]. Samples in routine alcohol analysis are recommended to be tested in duplicates. In postmortem toxicology, a BAC less than 0.1 g/L is usually reported as negative.

## 3. Higher Alcohols as Biomarkers of Postmortem Ethanol Production

The possible sources of higher alcohols in postmortem human specimens could be, firstly, the ante mortem ingestion of alcoholic beverages by a living person and, secondly, the postmortem microbial neoformation, either in corpse pre-sampling or in situ post-sampling. The detection of either higher alcohol in postmortem specimens could be due to either source alone, or due to a combination of both (as it is the case for ethanol). The postmortem higher alcohol detection due to ante mortem endogenous higher alcohol production, by microbial fermentation in the intestine (during the “auto-brewery” syndrome where ethanol is produced endogenously [4]), to our knowledge, is not reported in the literature; however, we are of opinion that, theoretically, it is possible to occur.

Most of the relevant studies have considered the presence of higher alcohols (mainly 1-propanol) as a qualitative [2,3,20,21,22,23,24,30,31], or semi-quantitative [15,16,19,29], indicator of postmortem microbial activity. The simultaneous presence of two or more higher alcohols in a postmortem blood sample is suggested as a warning flag that the ethanol origin of this sample should be questioned [3,18]. Specifically, the presence of 1-butanol, either alone or with 1-propanol, has been suggestive of postmortem microbial production of ethanol, in general, or, specifically, in drowned persons, respectively [30,31,32].

### 3.1. Higher Alcohols in Autopsy Cases 

Literature data on the concentrations of the higher alcohols in blood from postmortem cases, irrespectively of their origin (alcohol consumption or microbial production) is sparse, and reports are mainly on the 1-propanol concentrations. The reported relative data on the blood higher alcohols and ethanol concentrations for postmortem cases are summarized in Table 2. Considerable concentrations of higher alcohols have been reported in a sample set of 93 postmortem blood samples from respective autopsy cases, with BAC higher than 0.10 g/L [36]. It is worth mentioning that that 1-propanol was detected simultaneously with ethanol in most of the samples (only one case was negative for 1-propanol), while methyl-butanols (the sum of amyl- and isoamyl-alcohol) were detected in less samples.

In another report, concerning a suicide victim, femoral and cardiac blood was analyzed for ethanol and higher alcohols (Table 2) [39]. The case history of the deceased supported the ingestion of a large amount of a high-proof fruit distillate before death, the death of the victim was attributed to the toxicity of ethanol, and signs of putrefaction at autopsy were not reported. In this postmortem case, the most possible source of the detected higher alcohols should be the consumption of alcohol before death [39], although some microbial production could not be excluded, nor supported, with certainty.

Two more studies have reported blood ethanol and 1-propanol concentrations from postmortem cases (Table 2) [13,40]. The first study reported the blood ethanol and 1-propanol concentrations (among those of other volatiles) determined in autopsy cases from natural, violent, and unspecified causes of death [40]. As can be seen in Table 2, blood 1-propanol concentrations were higher in cases from undetermined causes of death than in cases from violent and sudden causes of death, respectively. Interestingly, the higher 1-propanol concentrations were detected in cases with putrefaction [40]. The second study reported blood ethanol and 1-propanol concentrations for 42 postmortem cases; 18 out of the 42 tested samples were negative for 1-propanol [13]. Both these studies [13,40] reported that ethanol or 1-propanol was not detected in all postmortem samples (as was reported in Reference [36], as well), nor in all cases with putrefaction, in agreement with our previously expressed consideration [41]. Finally, the study of Bonde suggested that a blood concentration of 1-butanol higher than 0.3 mg/L (0.03 mg/dL) is a biomarker of putrefaction (because of microbial action) [32].

### 3.2. Post-Sampling Production of Higher Alcohols

Many studies reported on the microbial ethanol generation post-sampling in postmortem samples [2,3,6]. On the contrary, the post-sampling higher alcohols generation in postmortem blood and the relevant measured concentrations have been reported only for 1-propanol, in only one study (Table 2) [42]. It is worth mentioning that the authors considered that the detected 1-propanol concentrations in their samples (up to 3 mg/dL) were negligible [42]. Nevertheless, in the view of the current review and other studies [13,33,34,35,36,40], the reported 1-propanol concentrations [42] are typical of microbial production. Theoretically, the post-sampling production of higher alcohols and ethanol are equally possible to occur, under certain circumstances since they have a microbial origin.

At this point, it should be mentioned that the increased temperature and the duration of storage are recognized as the premium influencing factors of the post-sampling alcohol production [45], besides the microbial species and microbial burden present in the specimen [46,47]. Furthermore, the presence, or not, of glucose in the sample is suggested to be the absolute determinator of post-sampling ethanol production [48]. Generally, the recommended precautions to decrease, or eliminate, the microbial activity post-sampling are the proper blood sampling in tubes with preservatives and the storage at appropriate freezing conditions (4 °C) [45,46].

### 3.3. Higher Alcohols in Blood of Living Individuals 

The higher alcohols, 1-propanol, isobutanol, 1-butanol, and 3-methyl-1-butanol, are among the primarily targeted congeners during the alcohol congener analysis workflow and evaluation [49,50]. Particularly, 1-propanol and isobutanol are considered the most important of all congeners in this methodology [25], while methyl-butanols are of minor importance due to the very low concentrations in blood after alcohol consumption [25,43].

In Table 2 are presented the ethanol and higher alcohols concentrations reported in two selected studies of the same research group, after consumption of alcoholic beverages [43,44], for comparison with the postmortem cases shown in Table 2. The ingestion of an alcoholic beverage “free” of 1-propanol, resulted in the detection of negligible 1-propanol concentration attributed by the authors to “endogenous” production [44]. However, to our knowledge, the literature does not support the microbial production of 1-propanol in the presence of ethanol, and, since the authors did not perform analysis for the presence of branched higher alcohols, or 1-butanol, in their samples, the possibility of “endogenous” production should be questioned and further investigated. Finally, the detection of 1-butanol was not reported in specimens from living individuals. It is worth mentioning that the reported concentration of 1-propanol, when the BAC was 1.22 g/L [43], was comparable to the respective levels reported for postmortem cases of natural causes of death (Table 2A) [40].

As could be concluded from Table 2, the reported higher alcohol levels in postmortem cases are significantly higher than the respective levels reported after alcohol consumption for ante mortem specimens. Generally, the microbial activity in a human corpse is expected to result in the formation of variable and unpredictable ethanol and higher alcohol patterns. The postmortem higher alcohols production would follow the ethanol production since the respective microbial biochemical pathways are interactive [38]. In addition, in postmortem cases with high BAC, it is possible that part or all the detected ethanol was due to ante mortem alcohol consumption, and the same could be true for the measured higher alcohol concentrations. Obviously, more data are needed on the abundance and corresponding concentrations of all higher alcohols, and especially the branched ones, in postmortem cases, in order to assess to which extent, the postmortem ethanol production is followed by higher alcohols’ production and what concentrations of higher alcohols could result from alcohol ingestion, or from postmortem production.

### 3.4. Specific Roles of 1-Propanol 

As was already mentioned, the higher alcohol mostly correlated to putrefaction and microbial ethanol production is 1-propanol [15,16,18,19]. Some studies have suggested the semi-quantitative approach of determination of the ratio of ethanol and 1-propanol concentrations in an autopsy specimen to distinguish the source of ethanol in the postmortem [15]. The authors proposed a ratio of ethanol to 1-propanol less than 20:1 in postmortem blood from rats’ carcasses, as suggestive of postmortem production [15]. Another study reported that when ethanol concentration in the brain was ≥0.50 mg/g with cerebral ethanol to 1-propanol ratio ≥40, and, when the concentration of ethanol is 0.10 to 0.49 mg/g with the ethanol to 1-propanol ratio ≥60, drinking should strongly be suspected [19]. However, other reports questioned or disproved the reliability of the ethanol to 1-propanol concentration ratio, mainly because 1-propanol was not detected in all postmortem cases where ethanol production was suspected [26,41]. Our previous studies [33,34,35,36] agree with those who questioned the reliability of the ethanol/1-propanol ratio per se, since it was shown that different bacteria could produce different patterns of 1-propanol, and other alcohols, in variable concentrations depending on the growth conditions (discussed in the next section).

More recently, another semi-quantitative approach, the 1-propanol concentration of 0.104 mg/dL (the “1-propanol criterion”), was suggested as an effective threshold concentration (“cut-off”) for “flagging” an autopsy blood sample as positive or “negative” for postmortem ethanol production [29]. It was proposed as a measurable laboratory indicator of microbial ethanol production in samples where 1-propanol and ethanol were detected. It could discriminate possible “positive” for postmortem ethanol production bloods from the “negative” ones, even in cases without obvious signs of putrefaction, such as corpses with external or internal traumatic lesions [29]. The authors suggested further tests to be performed on the “positives” to aid the final decision. This is an interesting approach which could be followed for other higher alcohols, as well, but it needs further testing in many samples, as well as case series, to prove its validity.

## 4. Modeling Microbial Ethanol Production

The approach of “modeling microbial (postmortem) ethanol production” has considered all the higher alcohols (1-propanol, 1-butanol, isobutanol, amyl- and isoamyl-alcohol) as quantitative biomarkers of postmortem ethanol production [33,34,35,36], beyond their previous recognition, as qualitative indicators of postmortem ethanol production.

### 4.1. Higher Alcohols in Microbial Cultures

The first stage of the modeling procedure was the development of bacterial, clostridial, and fungal cultures under controlled laboratory conditions, selected to approximate the conditions after death, and the recording of the concentrations of ethanol and higher alcohols produced by the microbes [33,34,35,36]. In Table 3 are presented the maxima ethanol and higher alcohol concentrations detected in the respective cultures. Apparently, ethanol was the most abundant alcohol produced by all microbes tested in laboratory cultures. Moreover, in all but one of the tested bacterial cultures, 1-propanol’s concentrations preceded 1-butanol’s or branched higher alcohols’ concentrations. The only exception was the *E. faecalis* cultures, where 1-butanol preceded the other higher alcohols. This observation probably explains why 1-propanol is the most studied higher alcohol in postmortem cases, given that bacteria are the most common inhabitants of a dead body and that they, usually, invade first the different body compartments after death [22,27,37,38].

On the other hand, the clostridia species (*C. perfrigens* and *C. sporogenes*) produced more 1-butanol than 1-propanol. This observation probably indicates that clostridia prefer the butyrate-butanol-acetate fermentation pathway, which produces 1-butanol directly from carbohydrates (Table 1), in parallel with the ethanol-producing pathways [38]. The predominant production of 1-butanol by clostridia (obligate anaerobes), which grow during late putrefaction, is in accordance with the studies that have correlated the presence of 1-butanol to putrefaction [32], or cases with oxygen deficiency [31].

Finally, it is worth mentioning that, in all *C. albicans* cultures, methyl-butanols’ levels (followed by isobutanol) predominated against 1-propanol’s levels, while 1-butanol levels were negligible. Therefore, it is reasonable to conclude that *C. albicans*, under the applied laboratory conditions, prefer the synthesis of branched higher alcohols than the neoformation of 1-propanol. This aspect differentiates fungal cultures [36] from the bacterial cultures [33,34,35] studied so far, where 1-propanol was the predominant alcohol, and from clostridial cultures [33], where 1-butanol predominates higher alcohols. (It should be underlined, at this point, that the reported methyl butanol levels in the respective studies [33,34,35,36] were the sum of amyl- and isoamyl-alcohols’ concentrations which were produced in equal amounts in all cultures.). The comparison of the higher alcohols levels determined in laboratory cultures, presented in Table 3, to those reported for postmortem cases, in Table 2, makes apparent some interesting facts. Firstly, the concentrations of the branched alcohols in the microbial cultures are comparable to the respective concentrations reported for postmortem cases, or even higher for certain microbes (*C. sporogenes*). Secondly, the concentrations of 1-butanol are higher in the cultures than in postmortem cases, while some postmortem cases have 1-butanol levels as high as those determined for the *C. sporogenes* cultures. Thirdly, 1-propanol concentrations in cultures are within the range of 1-propanol reported usually for postmortem cases (Table 2).

The initial glucose levels to the BHI and SDB culture media (used for the microbial cultures in Table 3) were 2 g/L and 20 g/L, respectively [33,34,35,36]. If assumed that all the glucose in each culture could ferment to ethanol, then the ethanol concentration should have been, theoretically, 1 g/L and 10 g/L in the BHI and the SDB cultures, respectively. However, the ethanol concentrations in microbial cultures (Table 3) allow the conclusion that the yield of ethanol neoformation ranged from 15% (in the BHI cultures of *C. perfrigens* and *E. faecalis*) to 100% (in the SDB cultures of *C. albicans*). It became apparent, as is already known [2,3], that the most influencing factors of the ethanol yield are the microbe species, the media composition (especially glucose availability), and the aeration conditions (anaerobic conditions by obligate anaerobic species seem to favor ethanol neoformation). Moreover, it was found that temperature affected ethanol and higher alcohols production in cultures of all tested microbes, under laboratory conditions [33,34,35,36]. It is reasonable to assume that the yields of neoformed ethanol in real cases would be within the respective range observed in laboratory cultures (15–100%), with the glucose levels and the microbe species being the main determinants of the final ethanol concentration and yield in each case.

### 4.2. Models to Calculate Microbial Ethanol

In the second stage of the modeling procedure, the experimental results on alcohols concentrations that were produced in cultures were statistically analyzed by stepwise multiple linear regression analysis. The process resulted in modeling the relationship between ethanol concentrations, as the dependent variable, in a linear correlation with the concentrations of the other alcohols, as the independent variables. Each independent variable was gradually introduced to build the model, allowing considerations for its significance to the final model and possible existing correlation with the other variables. The constructed models are first-degree mathematical equations for the calculation of the concentration of the microbially produced ethanol in a sample, as a function of the concentrations of higher alcohols determined in the same sample. Obviously, the constructed models constitute quantitative tools to estimate the microbially produced ethanol at postmortem specimes.

The microbial models reported so far and the laboratory conditions applied for their construction, as reported in the relevant studies, are summarized in Table 4. More than one model was constructed for all studied species, by skipping, gradually, the less significant variable (higher alcohol) and constructing a model with the remaining variables, except for *E. faecalis*. This procedure was followed for all datasets and in every case that a satisfactory model had resulted. Furthermore, four more models, two models that were constructed for *C. perfrigens*, one model for *E. coli*, and one model for *C. sporogenes*, as described previously [33], are included in Table 4 (indicated with an asterisk) and are reported herein for the first time. The linear correlation between higher alcohols and ethanol is more successful as the factor R^2^ increases.

The comparison of the models presented in Table 4 makes obvious that the significance of each higher alcohol as a descriptor of the relevant model (expressed by the factor before the respective higher alcohols concentration) is not analogous to its abundance, as expressed by the relative concentrations in the respective culture (the most abundant higher alcohol in the respective culture is shown in Table 4 in bold letters). For instance, the models constructed for *E. coli* under anaerobic conditions have 1-butanol as the most significant alcohol, followed by methyl-butanols, although 1-propanol was the most abundant higher alcohol in the respective cultures.

Overall, the modeling of PEP and the relevant models suggests that the higher alcohols, 1-propanol, 1-butanol, methyl-alcohol (both amyl- and isoamyl alcohol), and isobutanol, are quantitative indicators of microbial ethanol production. Furthermore, the quantitative relationships between the microbial higher alcohols’ concentrations and the microbial ethanol’ concentrations, as expressed by the suggested models, allow the estimation of the postmortem neoformed ethanol concentration

The applicability of the constructed models was tested in a series of autopsy blood samples where ethanol was detected simultaneously with at least one higher alcohol [33,34,35,36]. The application of each model in each case was considered successful if the estimated microbially produced ethanol, varied within an acceptable accuracy range from the measured BAC (usually E < ±40% or less).

The relevant studies reported that the bacterial models succeeded the predefined score (E < ±40%) in 19% of the cases for the *S. aureus* models [35], up to 45% of the cases the *E. coli* models [33,34]. The *C. sporogenes* and *C. perfrigens* models achieved the predefined score in 45% to 68% of the cases, respectively [33]. When each bacterial or clostridial model was applied to cases with BAC < 0.7 g/L, the predefined score was achieved in 68% of the cases by the *E. coli* models, and in 85% of the cases by the clostridia ones [33,34]. When the respective models were applied in cases with obvious signs of putrefaction, the *E. coli*, *E. faecalis*, *C. perfrigens*, and *C. sporogenes* models achieved the score in more than 95% of the cases [33,34,35]. Furthermore, the yeast models achieved the predefined score of applicability in only three percent of the cases [36]. Although the authors did not report any case characteristic of those autopsy cases, the result could indicate that the yeasts models could apply in specific cases where yeasts have grown in postmortem blood in the presence of elevated glucose levels. Finally, the authors tested the applicability of their models in denatured blood microbial cultures [35,36], in postmortem blood microbial cultures [34], and in blood products cultures [33] developed under various laboratory conditions. Overall, the relative results indicated that there is potential for application of the models in postmortem cases where higher alcohols have been produced simultaneously with ethanol.

## 5. Concluding Remarks

So far, the most studied and the most abundant higher alcohol in postmortem cases is 1-propanol. However, it is not the only higher alcohol which indicates microbial ethanol production. The five higher alcohols, 1-propanol, 1-butanol, isobutanol, 3-methyl-2-butanol (amyl-alcohol), and 2-methyl-1-butanol (isoamyl-alcohol), are qualitative and quantitative indicators of microbial ethanol production, as it is indicated by the respective bacterial, clostridial, and fungal ethanol production models.

The models of microbial ethanol production are tools offering additional data to properly interpret BAC in postmortem cases, as well as to define the origin of ethanol in the sample. The employed, relatively simple linear models for estimating microbial ethanol production provide a noteworthy accuracy in cultures and autopsy blood.

In the view of this review, the proper interpretation of postmortem ethanol analysis results should follow a step-by-step approach to estimate the suggested in the literature indicators, starting from the more easily obtainable, the detection of 1-propanol, isobutanol, methyl-butanols, and 1-butanol, during the chromatographic ethanol analysis. Then, the microbial models should be used to calculate the microbial generated ethanol and, subsequently, to evaluate the results in relation to case specific characteristics, such as putrefaction state, clinical history, and others. Finally, more selective analyses, providing more elaborate indicators of the ethanol origin (such as ethyl-glucuronide, ethyl-sulfate, and serotonin metabolites), could be performed, to provide complementary data that would aim accurate interpretation of postmortem BAC.

In conclusion, the evaluation of the postmortem BAC remains a complicated, multifunctional procedure. Needless to underline that more studies should be performed regarding the levels of higher alcohols in postmortem and ante mortem cases, to document the extent of applicability of the suggested models, to achieve feasible accuracy in interpreting the results of ethanol analysis, and to estimate the BAC origin in postmortem samples with increased certainty.

## Figures and Tables

**Table 1 molecules-27-00700-t001:** The biochemical pathways (fermentations) followed by the most common microbes activated at the postmortem to produce ethanol and higher alcohols [38].

Ethanol/Higher Alcohols	Biochemical Pathway	Microbe Domain
Ethanol	Glucose	Bacteria, Clostridia, Yeasts
	Glycerol	Clostridia, Bacteria, Yeasts
	Mixed acid and butanediol	Bacteria
/1-Butanol	Butanol-acetone and Butyrate	Clostridia, Bacteria
	Glycerol	Clostridia, Bacteria
/Amyl- and Isoamyl-alcohol	Amino acids (linked to pyruvate availability)	Bacteria, Clostridia Yeasts,
/Isobutanol	Amino acids (linked to pyruvate availability)	Bacteria, Clostridia, Yeast
/1-Propanol	Acetone (from pyruvate, FA, Glycerol)	Bacteria, Clostridia
	Amino acids (linked to pyruvate availability)	Bacteria, Clostridia, Yeasts
	Glycerol	Bacteria, Clostridia, Yeasts

**Table 2 molecules-27-00700-t002:** Ethanol and higher alcohols detected in postmortem blood from autopsy cases (A), after storage post-sampling (B), and in plasma or serum from living persons (C).

Sample Origin	Specimen (N)	C_max_ Ethanol, g/L	C_max_ Higher Alcohols, mg/dL				Ref.
			1-Propanol	1-Butanol	Isobutanol	Methyl-Butanol (Amyl/Isoamyl Alcohol)	
(A) Postmortem	Blood (93)	0.10–4.55	13.62	12.32	1.85	0.48 ^1^	[36]
	Blood, Femoral (1)	6.62	2.4	0.05	0.45	1.26 ^1^ (0.28/0.98)	[39]
	Blood, Cardiac (1)	8.11	2.3	0.04	0.51	1.20 ^1^ (0.28/0.92)	[39]
	Blood, Natural COD * (212)	6.01	0.18	nd	nd	nd	[40]
	Blood, Violent COD (243)	6.02	12.0	nd	nd	nd	[40]
	Blood, Undetermined COD (28)	2.68	32.5	nd	nd	nd	[40]
	Blood (42)	0.07–4.64	7.0	nd	nd	nd	[13]
	Blood (1)	0.97	8.6	nd	nd	nd	[6]
	Blood	-	-	>0.03	-	-	[32]
(B) Postmortem/ Post sampling	Blood (1)	0.59/4.9	~0.2/0.4	nd	nd	nd	[42]
	Blood (1)	2.1/9.6	~0.4/3.0	nd	nd	nd	[42]
(C) Antemortem	Plasma	0.84/1.22	0.042/0.29	-	0.03/0.09	0.04 ^1^ (0/0.04)	[43]
	Serum	0.65–1.23	<0.03	nd	nd	nd	[44]

^1^ the reported values represent the sum of the concentrations of amyl- and isoamyl-alcohols determined in the samples. * COD: cause of death. nd: not determined.

**Table 3 molecules-27-00700-t003:** Ethanol and higher alcohols detected in microbial cultures under laboratory conditions.

Microbe Domain	Microbe, Culture Conditions	Max Ethanol, g/L	Max Higher Alcohols, mg/dL				Ref.
			1-Propanol	1-Butanol	Isobutanol	Methyl-Butanol ^1^ (Amyl/Isoamyl Alcohol)	
(A) Bacteria	*E. faecalis*, BHI—Ae/An	0.15	0.10	0.19	0.01	0.03	[35]
	*S. aureus*, BHI—Ae/An	0.28	0.20	0.14	0.08	0.13	[35]
	*S. aureus*, BHI—An	0.36	0.23	0.15	0.08	0.11	[35]
	*K. pneumoniae*, BHI—Ae/An	0.60	2.78	0.10	0.10	1.08	[35]
	*K. pneumoniae*, BHI—An	0.64	2.78	0.14	0.07	0.72	[35]
	*E. coli*, BHI—An	0.56	1.02	0.16	0.10	0.12	[33]
	*E. coli*, BHI—Ae	0.55	2.08	0.21	0.02	0.05	[34]
	*E. coli*, BHI—Ae/An	0.62	3.63	0.24	0.03	0.11	[34]
(B) Clostridia	*C. perfrigens*, BHI—An	0.15	0.70	1.90	0.15	0.10	[33]
	*C. sporogenes*, BHI—An	0.87	10.2	11.9	6.10	0.57	[33]
(C) Yeasts	*C. albicans*, BHI—Ae/An	0.62	0.102	0.05	0.33	1.30 *	[36]
	*C. albicans*, BHI—An	0.89	0.12	0.06	0.45	1.48 *	[36]
	*C. alcicans*, SDB—Ae/An	9.83	0.51	0.18	0.85	1.71 *	[36]
	*C. alcicans*, SDB—An	10.1	0.46	0.15	0.82	1.70 *	[36]

^1^ the reported values represent the sum of the concentrations of amyl- and isoamyl-alcohols determined in the samples. * Equal concentrations of amyl- and isoamyl-alcohols determined [35]. BHI: Brain Heart Infusion. SDB: Sabouraud Dextrose Broth. Ae: aerobic conditions. An: anaerobic conditions.

**Table 4 molecules-27-00700-t004:** The microbial models allowing to calculate the microbial generated ethanol concentration (in g/L) in a postmortem blood from the concentrations of the higher alcohols (in mg/dL) being detected in the same sample, the respective factor, and characteristics (microbe, aeration, culture medium) of the cultivation conditions used for their constructions.

Microbe Domain	No	Model (Equation)	R^2^	Microbe-Aeration Conditions, Culture Medium	Ref.
**Bacteria**	1	Ethanol = 0.16 * 1Propanol − 1.24 * Isobutanol + 0.27 * **1Butanol** + 0.09	0.37	*E. faecalis*-Ae/An	[35]
	2	Ethanol = 0.28 * **1Propanol** + 3.52 * Isobutanol + 0.91 * 1Butanol − 1.05 * Methyl-butanol	0.85	*S aureus*-Ae/An	[35]
	3	Ethanol = 0.40 * **1Propanol** + 5.33 × Isobutanol + 0.04 × 1Butanol − 1.65 × Methyl-butanol + 0.01	0.91	*S. aureus*-An	[35]
	4	Ethanol = 0.13 * **1Propanol** + 6.17 * Isobutanol + 1.37 * 1Butanol − 0.43 * Methyl-butanol − 0.02	0.94	*K. pneumoniae*-Ae/An	[35]
	5	Ethanol = 0.31 * **1Propanol** − 0.41 * Methyl-butanol + 0.05	0.88	*K. pneumoniae*-Ae/An	[35]
	6	Ethanol = 0.23 * **1Propanol** + 1.20 * 1Butanol − 0.27 * Methyl-butanol + 0.03	0.90	*K. pneumoniae*-An	[35]
	7	Ethanol = 0.36* **1Propanol** − 0.71 * Methyl-butanol + 0.10	0.83	*K. pneumoniae*-An	[35]
	8	Ethanol = 0.07 * **1Propanol** + 0.20 * Isobutanol + 1.61 * 1Butanol + 1.15 * Methyl-butanol + 0.15	0.75	*E.coli*-An	[33]
	9 ^a^	Ethanol = 0.08 * **1Propanol** + 1.57 * 1Butanol + 1.18 * Methyl-butanol + 0.15	0.74	*E.coli*-An	[33]
	10	Ethanol = 2.25 * 1Butanol + 0.98 * Methyl-butanol + 0.15	0.72	*E.coli*-An	[33]
	11	Ethanol = 0.07 * **1Propanol** + 3.77 * Isobutanol − 0.26 * 1Butanol + 0.07 * Methyl-butanol + 0.39	0.83	*E.coli*-Ae	[34]
	12	Ethanol = 0.05 * **1Propanol** + 1.13 * Isobutanol + 0.95 * 1Butanol − 1.60 * Methyl-butanol + 0.31	0.90	*E.coli*-Ae/An	[34]
	13	Ethanol = 0.05 * **1Propanol** + 0.53 * 1Butanol + 0.32	0.75	*E.coli*-Ae/An	[34]
	14	Ethanol = 0.05 * **1Propanol** + 1.06 * Isobutanol + 1.76 * 1Butanol − 1.62 * Methyl-butanol + 0.14	0.81	*E.coli*-An	[34]
**Clostridia**	15	Ethanol = 0.08 * 1Propanol + 0.03 * **1Butanol** + 0.30 * Isobutanol − 0.01 * Methyl-butanol + 0.03	0.94	*C. perfringens*-An	[33]
	16 ^a^	Ethanol = 0.11 * 1Propanol + 0.03 * **1Butanol** + 0.13 * Isobutanol + 0.03	0.96	*C. perfringens*-An	[33]
	17 ^a^	Ethanol = 0.13 * 1Propanol + 0.03 * **1Butanol** + 0.03	0.94	*C. perfringens*-An	[33]
	18	Ethanol = − 0.11 + 4.71 * 1Propanol	0.92	*C. perfringens*-An	[33]
	19	Ethanol = 0.16 * 1Propanol − 0.07 * **1Butanol** + 0.61 * Methyl-butanol − 0.07 * Isobutanol + 0.05	0.64	*C. sporogenes*-An	[33]
	20	Ethanol = 0.17 * 1Propanol − 0.03 * **1Butanol** − 0.11 * Isobutanol + 0.06	0.62	*C. sporogenes*-An	[33]
	21	Ethanol = 0.15 * 1Propanol − 0.14 * Isobutanol + 0.09	0.60	*C. sporogenes*-An	[33]
**Yeast**	22	Ethanol = 3.01 × 1Propanol − 0.09 × **Methyl-butanol** + 0.46 × **Isobutanol** + 0.28	0.49	*C. albicans*-Ae/An, BHI	[36]
	23	Ethanol = 3.98 × 1Propanol − 0.25 × **Methyl-butanol** + 1.10 × **Isobutanol** + 0.33	0.68	*C. albicans*-An, BHI	[36]
	24	Ethanol = 4.30 × 1Propanol + 0.29 × **Isobutanol** **(or Methyl butanol)** + 0.28	0.67	*C. albicans*-An, BHI	[36]
	25	Ethanol = 10.4 × 1Propanol − 2.24 × **Methyl-butanol** + 9.62 × **Isobutanol** + 0.76	0.95	*C. albicans*-Ae/An, SDB	[36]
	26	Ethanol = 10.4 × 1Propanol + 5.58 × **Isobutanol** (or **Methyl-butanol**) + 0.43	0.95	*C. albicans*-Ae/An, SDB	[36]
	27	Ethanol = 20.6 × 1Propanol + 4.13 × **Methyl-butanol** − 5.16 × **Isobutanol** − 0.42	0.95	*C. albicans*-An, SDB	[36]
	28	Ethanol = 21.5 × 1Propanol + 1.31 × **Isobutanol** (or **Methyl-butanol**) − 0.38	0.95	*C. albicans*-An, SDB	[36]

^a^ These models are presented here for the first time, although they were constructed previously, as described in the respective references.

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
