# Peer review of "Modeling Postmortem Ethanol Production/Insights into the Origin of Higher Alcohols"

_molecules, 2022, doi:10.3390/molecules27030700_

Round 1

Reviewer 1 Report

This review manuscript summarizes the role of fatty alcohols as biomarkers of postmortem microbial ethanol production, and more specifically, the modeling of postmortem ethanol production.

The content was interesting, but I thought it might be difficult for readers to understand because it was not divided into detailed chapters. Please make the chapters more detailed so that the readers can easily understand.

In particular, I think that "2. Higher alcohols as biomarkers of postmortem ethanol production" and "3. Modeling microbial (postmortem) ethanol production" need to be described in more detail.

In addition, it would be good to insert diagrams that are easy to understand visually. The following is a detailed revision.

Line 63 posmortem → postmortem

Line 240 presnece → presenece

Reviewer 2 Report

The focus of this review is on the current status and avenues for building scientific models to distinguish between alcohol levels at the time of death and the source of alcohol measured in postmortem blood, specifically pre-mortem ingestion or microbial production of alcohol, and to properly interpret the associated BAC results. In particular, the role of higher alcohols as biomarkers of postmortem microbial ethanol production, and more specifically, the modeling of postmortem ethanol production, has become an important approach for making inferences in criminal science to unravel the factors of a case, and the review in this special issue In my opinion, some of the important information is provided by this review article. In my opinion, after correction of the following errors, I strongly recommend that the article be published.

Correction points

misspell: Page 6, line 240

“presnece” to “presence”

misspell:Page 2, line 97

Embden-Meyerhof-Parnas (EMD) “glucolytic” pathway to “Embden-Meyerhof-Parnas (EMD) “glycolytic” pathway” 

misspell:Page  1, line 14, Page 10, line 385

“isoamyl-allcohol” to “isoamyl-alcohol”

misspell:Page 2, line 63

“posmortem” to “postmortem”

missing word:Page 9, line 362

of the cases the S. aureus models, to of the cases “for” the S. aureus models

misspell:Page 9, line 364, 368

  1. perfrigens, to “C. perfringens”, line 364, 368

Reviewer 3 Report

The review article under the title "Modeling postmortem ethanol production / Insights into the origin of higher alcohols" is a good attempt with few minor errors and typos. The author is expert in the title field and the review article covers the field of his expertise. The following errors are pointed out and the author is supposed to rectify them before final decision. 

  1. There are some inconsistencies in the text which need to be rectified such as "Ante mortem and antemortem"; "postmortem and post mortem"; and "neo formed and neoformed" etc.  
  2. some of the statements need to be polished to become clear, such as, line 49: "as regards ethanol levels......"; Line 76: "production concerned the  bacteria"; Line 83: "is a complicate process"; Line 168: replace mg/l by mg/L; Line 175: "and concern only ....."; Line 178: "considered negligible the detected......."; Line 187: insert degree(not zero as superscript, 0) properly from symbols; Line 204: check punctuations "it is possible part, or all, of ........."; Line 242: "external of internal"
  3. Line 51: [2-3, 5-6, the square bracket at the end is missing and also instead of hyphen there must be a comma, the same may be applied to the entire text.  check references in line 71 and so on. 
  4. This reviewer suggests to add a section related to analytical methods/sensors and their limitations. It will enhance the importance of the review and will attract the  scientific community to a greater extent.  
